# Efficacy of adding activity of daily living simulation training to traditional pulmonary rehabilitation on dyspnea and health-related quality-of-life

**Kayla Mahoney**[1], **Jacqueline Pierce**[1], **Stacey Papo**[2], **Hafiz Imran**[1,3], **Samuel Evans**[2], **Wen-Chih Wu**[1,3] *

**1** The Miriam Hospital Center for Cardiovascular and Pulmonary Rehabilitation, Providence, RI, United States of America, **2** The Newport Hospital Cardiovascular and Pulmonary Rehabilitation, Newport, RI, United States of America, **3** Providence VA Medical Center, Providence, RI, United States of America

\* wen-chih_wu@brown.edu

## Abstract

### Introduction

Exercise modalities offered as part of traditional pulmonary rehabilitation (PR) do not always translate to successful performance of Activities of Daily Living (ADL) and may hinder gains in patient's sense of well-being. Data is lacking on the efficacy of incorporation of ADL-focused training in PR. The aim of this study was to determine the impact of incorporation of ADL simulation and energy-conservation training in PR as part of a quality-initiative on health-related-quality-of-life (HRQOL), dyspnea, fatigue, and six-minute-walk-test among PR patients.

### Methods

Retrospective study where medical records of consecutive patients with chronic respiratory diseases who completed PR from 2016 to 2018 were reviewed. ADL-focused energy-conservation training was added to traditional PR in September 2017 by replacing three monthly sessions of traditional PR with energy-conservation training as a quality-improvement-initiative. The change from baseline on HRQOL measured by COPD assessment test (CAT), six-minute-walk-test, MMRC dyspnea score and CRQ-dyspnea and CRQ-fatigue questionnaires, were compared between patients who received traditional PR versus energy-conservation PR. Within and between group differences were calculated via repeated-measures ANOVA.

### Results

The baseline characteristics of 91 patients who participated in traditional PR versus energy-conservation PR (n = 85) were similar (mean age = 68.6±10.4 years, 49% men). While improvement from baseline was similar and significant for both groups for MMRC, CRQ-dyspnea and CRQ-fatigue scores, and six-minute walk test, patients who participated in

**Data Availability Statement:** Data cannot be shared publicly because of HIPAA regulations. A de-identified copy of the Data are available from the Miriam Hospital Institutional Ethics Committee

(contact via email to Adrienne McParlin: amcparlin@lifespan.org) for researchers who meet the criteria for access to confidential data.

**Funding:** The author(s) received no specific funding for this work.

**Competing interests:** The authors have declared that no competing interests exist.

energy-conservation PR had significantly higher improvement in HRQOL CAT scores (p = 0.01) than those who completed traditional PR.

## Conclusion

Tailoring patient's training programs to include energy-conservation training exercises specific to ADL in PR improved HRQOL over traditional PR in patients with chronic respiratory diseases despite no significant change in functional status. Future randomized-controlled trials will be needed to confirm these initial findings.

## Introduction

Pulmonary Rehabilitation (PR) is a multi-faceted intervention offered to patients with chronic respiratory diseases that includes patient tailored exercise programming and education aimed at promoting positive health behavioral changes for the self-management of disease [1–4]. Pulmonary rehabilitation has been demonstrated in patients with respiratory diseases of varying severity to improve exercise tolerance, health-related-quality-of-life (HRQOL), dyspnea and fatigue, utilization of medical therapies, reducing hospitalizations, and improving survival [1–5]. However, patients with respiratory diseases may still find difficulties in performing their activities of daily living (ADLs) which may attenuate the potential benefits achieved in their sense of well-being and/or functional status.

Evidence based clinical guidelines for exercise prescription have been published by two leading organizations including the American Thoracic Society and the American Association of Cardiovascular and Pulmonary Rehabilitation for improving dyspnea, functional capacity and HRQOL among patients with chronic obstructive pulmonary disease (COPD) [2, 3]. Both organizations agree that patients should participate in 4 to 12 weeks of supervised light to moderate intensity cardiovascular exercise 3 to 5 times per week for 20 to 60 minutes per session using walking and or cycling as primary exercise modalities. Guidelines for resistance training exercise have also been developed but lack specificity. While it is well understood that both upper and lower extremity resistance training exercises be included as part of a balanced exercise program in PR, data on tailored training specific to improving performance with ADLs, energy-conservation strategies and its effects on outcomes remain lacking in PR.

In addition to following the guidelines for exercise prescription, clinicians should also consider inclusion of energy-conservation techniques in all PR based on The American Association of Cardiovascular and Pulmonary Rehabilitation guidelines [3]. Energy conservation are techniques often used by occupational therapy focused on pacing, posture, and breathing to reduce the physical demand of common activities that individuals find challenging and or to meet occupational needs [6, 7]. These techniques include use of pursed lips breathing and avoidance of forward bending since they have been shown to reduce the dynamic hyperinflation experienced by COPD patients during performance of certain ADLs [7]. Learning and applying energy conservation techniques to reduce patients' energy expenditure and dyspnea with ADL's have the potential to improve functional performance and quality of life in COPD patients [7, 8].

Previous study on individualized occupational therapy focused on energy conservation did not improve occupational performance or satisfaction over usual care in patients with COPD [8], but formal PR was not part of the intervention. When PR was combined with the application of energy conservation techniques among patients with COPD, some investigators

observed improvement in time to perform ADL tests, functional capacity, and self-reported Borg symptom ratings of dyspnea and fatigue [9]. However, a comprehensive assessment of the patient's HRQOL as result of these changes has not been performed [10]. Moreover, the efficacy of energy conservation training as part of a comprehensive PR in non-COPD patients as well as its comparison with traditional PR without energy conservation training are both unknown. The aim of this retrospective study based on the implementation of a quality initiative was to investigate the change in HRQOL, fatigue, dyspnea, and six-minute walking distance before and after the incorporation of energy conservation in ADL training to traditional PR among patients with COPD and non-COPD chronic respiratory diseases enrolled in two PR programs.

## Methods

### Study design

The Miriam Hospital IRB approved this retrospective study under the title of "Patient characteristics, cardiac rehabilitation intervention and their relationship with cardiovascular outcomes", IRB# 637472, IRB Board #216514. The study has waiver of informed consent. The study was a retrospective, observational comparison of patients enrolled in PR before (October 2016 through August 2017) and after (September 2017 to September 2018) the addition of the new energy conservation in ADL simulation training to traditional PR intervention, as part of an institutional quality-improvement initiative in September of 2017.

### Study sample

Medical records of 176 patients with qualifying diagnoses for PR of COPD, asthma, pulmonary fibrosis, interstitial lung disease, lung transplant, lung cancer, bronchiectasis, pulmonary hypertension, restrictive lung disease, and chronic respiratory failure who completed PR in either a University Teaching Hospital or a Community Hospital were reviewed. Patients were not eligible for PR enrollment if they had unstable cardiac disease, uncontrolled diabetes or hypertension, or significant cognitive impairment. For this study, we only reviewed the records of patients who completed the 12 weeks of the PR program since patients who did not complete PR did not have the post-PR questionnaires and assessments.

All patients who enrolled in PR irrespective of the study period, completed the COPD Assessment Tool (CAT) for HRQOL [11], Chronic Respiratory Questionnaire (CRQ-dyspnea) and Modified Medical Research Council Questionnaire (MMRC) [12] for the assessments of dyspnea, and Chronic Respiratory Questionnaire (CRQ-fatigue) [13] for evaluation of fatigue, at the PR enrollment visit and at discharge from the program.

### Primary outcome

Our primary outcome was *Health Related Quality of Life or HRQOL,* as measured by CAT, since it encompassed a spectrum of domains that determined a patient's holistic self-sense of well-being [10]. The CAT questionnaire is both a reliable and valid tool for evaluating HRQOL among patients with COPD and is recommended by the American Thoracic Society for evaluating the impact of COPD on health status [11]. The CAT assesses cough, sputum, dyspnea, chest tightness, and sleep with each item scaled between one to five. Scores range between zero and forty where higher scores reflect greater impact of COPD severity on HRQOL. The American Association of Cardiovascular and Pulmonary Rehabilitation guidelines indicate that a change in CAT score $\geq 2$ in the negative direction as significant.

## Secondary outcomes

The *CRQ questionnaire* is also approved as a reliable and valid measure of HRQOL among COPD patients evaluating four domains of disease including dyspnea, fatigue, emotional, and mastery [13]. This study collected data on the dyspnea and fatigue domains where $\geq 0.5$ point improvement in individual scores has been determined as significant by the American Association of Cardiovascular and Pulmonary Rehabilitation. The CRQ dyspnea domain is particularly valuable as patients rate breathlessness on their five most important ADLs chosen from a list and is therefore unique to each individual [10]. The domain of fatigue was also considered since it is a common complaint of patients suffering from chronic respiratory illnesses [14].

The MMRC has been found to be a reliable and valid tool for the assessment of dyspnea and disability among the COPD patient population [12]. It is unique in our study population as it collects data on dyspnea with ADLs. Disease severity is graded zero to four where higher values correlate with greater dyspnea with ADLs.

**Exercise tolerance.** A six-minute walk test was performed both at baseline and discharge from PR to assess each patient's functional capacity [15]. Testing conditions were the same for both tests regarding the use of assistive devices, oxygen titration, and method for carrying oxygen. Absolute and relative contraindications to exercise testing defined by the American Thoracic Society guidelines were reviewed prior to administration of each walk test and same protocols were followed for consistency [15].

## Comparison groups

**Pre-implementation: Traditional Pulmonary Rehabilitation group (Traditional PR).** Patients attended 12 weeks of twice weekly supervised standard PR that consisted of exercise sessions for 60 minutes in addition to a 30-minute education class on exercise, nutrition, psychosocial components, medications, anatomy and physiology as it relates to breathing and lung disease, and symptoms. Exercise consisted of a combination of both individualized aerobic (examples: treadmill, arm ergometer, stationary bicycle, and NuStep) and resistance training modalities (examples: Keiser machines, light hand weights and resistance bands). No energy conservation training in ADL simulation scenarios were provided.

**Implementation: Energy Conservation Pulmonary Rehabilitation group (Energy Conservation PR).** The energy conservation PR consisted of same standard 12 weeks of twice weekly traditional PR exercise and education sessions, but one time per month (usually the second or third Tuesday of every month), energy conservation on ADL simulation training occurred in place of one traditional PR session. Session duration remained the same at 90 minutes and consisted of 60 minutes of targeted resistance training and energy conservation techniques focused on the performance of simulated ADLs, in addition to a 30-minute education session; hence equating the total exercise and education time from the traditional PR. Therefore, patients would have up to three separate occasions to receive energy conservation training throughout their 12-week program regardless of the timing of their enrollment as both facilities practice rolling enrollment. This energy conservation addition to the traditional PR program (i.e. energy conservation PR) resulted as part of an institutional quality-improvement initiative in September of 2017.

As part of the individualized energy conservation training, a baseline questionnaire (Self-Reported Task Difficulty, Fig 1) was administered prior to the exercise training. The Self-Reported Task Difficulty was extracted from a portion of the CAT and adopted to quantify the difficulty in performance of specific ADLs for training purposes. It is not yet validated amongst the non-COPD population. Patients rated difficulty of performance from zero (no difficulty) to five (extreme difficulty) on each individual ADL task including stairclimbing, bending,

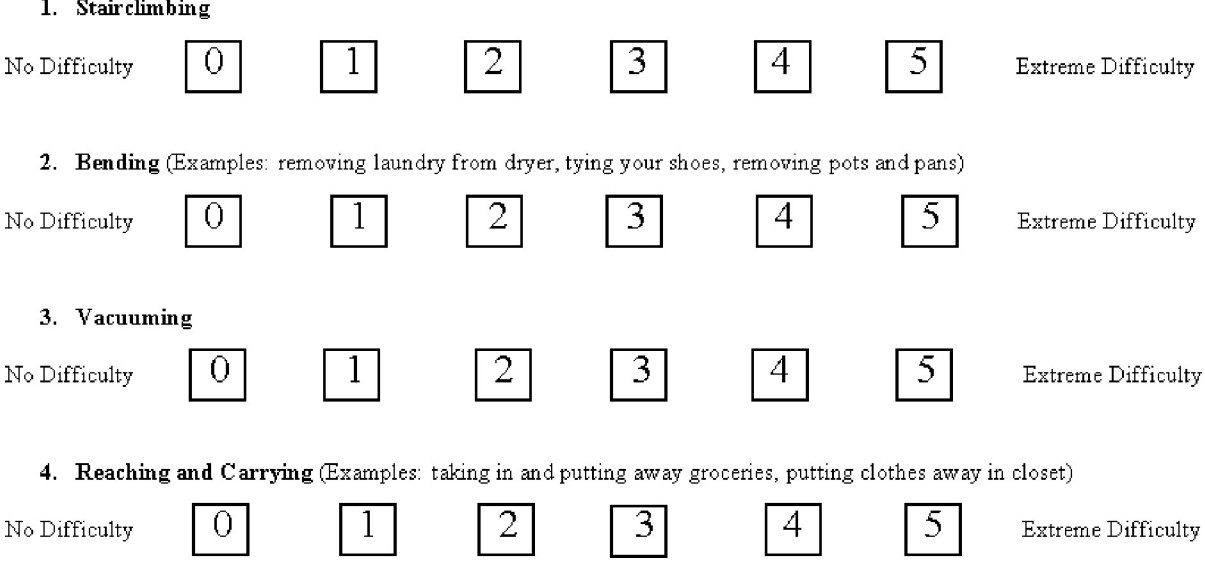

**Fig 1. Self-reported ADL task difficulty survey.**

reaching, and vacuuming. These tasks were then used as targets for the energy conservation training in ADL simulation workstations.

The energy conservation on ADL simulation training began with 15 minutes of an instructor led warm up which included coaching on posture and pursed lip breathing in combination with 5 repetitions of 10 upper and lower extremity exercises specific to the selected ADL tasks that the individuals find difficult to perform. Patients began seated performing body weight side bending, overhead reaching, toe touching (dynamic hamstring stretch), and leg extensions. Patients then performed body weight standing exercises using a chair for stabilization including bilateral hip hinge, split stance hip hinge, high knee marching, hamstring curls, and sit to stand exercises. Following the warm-up exercises, patients rotated through 12 supervised ADL simulation stations tailored to include exercises specific to ADLs that the patients had rated as difficult, such as stair-climbing, vacuuming, reaching and bending tasks. Stair-climbing specific stations included a single stair step up and or climbing a single flight of stairs with a staff member. Vacuuming specific stations included a dumbbell front raise, Theraband® CLX one arm row, Theraband® CLX one arm shoulder extension, and weight shift with reach simulation. Stations specific to reaching included bodyweight or dumbbell shoulder press, door frame diagonal (low to high unilateral trunk rotation), chopping (high to low bilateral trunk rotation, and carrying grocery bags and unloading items into an upper level kitchen cabinet). Bending stations included a laundry basket deadlift, and removing pots and pans from a lower level kitchen cabinet. In the act of performing these simulated ADL tasks, patients were instructed on energy conservation techniques including pacing, posture, pursed lip breathing and occupational considerations, and were provided with adapted modifications for each ADL task when appropriate. In addition, patients received a detailed handout of the ADL exercises and were encouraged to practice the exercises at home to increase the benefit of the program [16]. Patients were held accountable to performing the handout exercises at home by self-report on each of their regularly scheduled PR exercise sessions. The 30-minute education

class focused on providing patients with energy conservation strategies aimed at reducing the physical demand of common household activities including review on pacing, posture, and pursed lips breathing in addition to use of assistive devices, planning, prioritizing, and recruiting help from others when appropriate.

## Statistical analysis

Continuous variables are expressed as mean ± standard deviation (SD), and categorical variables in percentages (%). The differences in baseline characteristics between the patients who dropped-out versus those who completed PR, and between traditional PR and energy conservation PR groups were compared using T-tests for continuous and chi$^2$ for categorical variables. Change from baseline within the group and between the groups in primary and secondary outcomes were compared using repeated-measures ANOVA and Cohen's d was calculated for between group comparisons for each outcome. Given that PR encompassed a heterogeneous group of patients, to assess whether study findings vary by COPD status, we tested for multiplicative interaction between type of PR and COPD diagnosis using linear regression where the dependent variable was change from baseline for each of the study outcomes. In addition, exploratory analyses were conducted within the energy conservation PR group on the change over time in the patient self-rating of ADL task difficulty. Analyses was performed only on the available data without imputation of missing data. Statistical analysis was performed using statistical software (STATA SE version15.0). A two-sided P value of ≤0.05 is considered significant.

## Results

Out of the 142 patients enrolled between October 2016 and August 2017 (pre-implementation period), 51 (35.9%) patients did not complete the program. These patients were similar in age, gender and proportion of COPD diagnosis to the patients who completed the program (S1 Table). Out of the 138 patients enrolled between September 2017 and September 2018 (implementation of the energy conservation PR), 53 (38.4%) patients did not complete the program. Patients who dropped-out were on average 4 years younger but otherwise similar in gender and proportion of COPD diagnosis to the patients who completed the program (S1 Table). For both pre- and implementation periods, the top three reasons for dropped-out included medical event that precluded continued participation, non-compliance, and personal reasons.

Medical records of 176 patients (age 68.6 ± 10.4 years, FEV1 = 57.8 ± 26.3% predicted, FEV1/FVC = 0.6 ± 0.2%, 49% male) who completed the 12-week PR program were reviewed and analyzed. There were no significant differences in baseline characteristics between the Energy Conservation–PR (n = 85, 69% COPD and 31% non-COPD) and Traditional PR (n = 91, 59% COPD and 41% non-COPD) groups in age, gender, pulmonary function, functional status and pulmonary disease diagnoses (Table 1).

Results after the energy conservation PR and traditional PR interventions are summarized in Table 2. Baseline values of the questionnaires between energy conservation and traditional PR were similar (all P values >0.05). When compared to baseline, both groups demonstrated significant and similar improvements in the dyspnea score as measured by MMRC and CRQ-dyspnea scores, perception of fatigue (CRQ-Fatigue), and functional status assessed by the six-minute-walk test (all P values <0.01). For HRQOL as assessed by the CAT score, only patients in the energy conservation, but not the traditional PR group, had significant improvements from baseline (P <0.01), and it was of greater magnitude compared to the traditional PR group (P = 0.01 between groups). Subgroup analysis by COPD status was described in S2 Table. The change from baseline in CAT scores was significantly greater for the energy

**Table 1. Baseline characteristics.**

| Characteristics | Energy Conservation PR (n = 85) | Traditional PR (n = 90) | p-value |
|---|---|---|---|
| Caucasian (%) | 93 | 88 | 0.26 |
| Men (%) | 49.4 | 47.8 | 0.83 |
| Age, years (mean ± SD) | 70.0 ± 10.7 | 67.2 ± 9.9 | 0.08 |
| FEV$_1$, % predicted (mean ± SD)* | 56.8 ± 27.7 | 58.8 ± 25.0 | 0.62 |
| FEV$_1$/FVC, % (mean ± SD)* | 0.6 ± 0.2 | 0.6 ± 0.2 | 0.39 |
| BMI, kg/m$^2$ (mean ± SD)* | 29.3 ± 6.1 | 30.7 ± 7.2 | 0.16 |
| **Diagnosis (%)** | | | |
| COPD | 59 | 54 | 0.42 |
| Asthma | 5 | 8 | 0.63 |
| Pulmonary Fibrosis | 5 | 8 | 0.63 |
| Interstitial Lung Disease | 11 | 10 | 0.88 |
| Bronchiectasis | 5 | 2 | 0.36 |
| Lung Cancer | 4 | 3 | 0.93 |
| Pulmonary Hypertension | 11 | 10 | 0.88 |
| Lung Transplant | 0 | 2 | 0.17 |
| Cystic Fibrosis | 0 | 1 | 0.33 |
| Restrictive Lung Disease | 0 | 1 | 0.33 |
| Chronic Respiratory Failure | 0 | 1 | 0.33 |

**Abbreviations**: PR = Pulmonary Rehabilitation, TPR = Traditional Pulmonary Rehab, FEV = Forced Expiratory Volume, FVC = Forced Vital Capacity, BMI = Body Mass Index, 6MWT = Six-Minute Walk Test.

*Sample size Energy Conservation/Traditional PR groups: FEV1: n = 80/86; FEV1/FVC: n = 78/86; BMI: 85/89.

conservation versus traditional PR in the COPD group (p = 0.03) and trended towards significance in the non-COPD group (p = 0.06). The change from baseline in the remaining outcomes (CRQ-dyspnea, CRQ-Fatigue, MMRC, six-minute-walk test) was similar between the energy conservation and the traditional PR groups in both the COPD and non-COPD patients. Regression analyses using multiplicative interaction did not show any of the above results to significantly differ by COPD status (all P values >0.25).

**Table 2. Comparison between energy conservation and traditional pulmonary rehabilitation on patient outcomes.**

| | Energy Conservation PR (mean ± SD) | | | Traditional PR (mean ± SD) | | | Cohen's delta on the difference in change from baseline between Energy Conservation vs. Traditional PR (95% CI) | P-value On Change Between Groups by ANOVA |
|---|---|---|---|---|---|---|---|---|
| | n | Before PR | After PR | n | Before PR | After PR | | |
| **CAT** | 84 | 18.49 ± 7.17 | 14.75 ± 7.19 | 90 | 17.38 ± 6.79 | 16.76 ± 7.50* | -0.43 (-0.74 to -0.14) | 0.004 |
| **CRQ Dyspnea** | 84 | 15.93 ± 4.77 | 20.67 ± 6.06 | 90 | 14.90 ± 5.34 | 18.04 ± 6.51 | 0.23 (-0.06 to 0.53) | 0.12 |
| **CRQ Fatigue** | 84 | 15.33 ± 4.72 | 18.27 ± 4.64 | 90 | 14.77 ± 4.08 | 17.10 ± 4.05 | 0.13 (-0.17 to 0.43) | 0.38 |
| **MMRC** | 85 | 1.93 ± 0.99 | 1.33 ± 0.88 | 90 | 1.80 ± 0.90 | 1.49 ± 1.00 | 0.22 (-0.08 to 0.52) | 0.10 |
| **6MWT** | 80 | 315.69 ± 114.73 | 363.62 ± 118.74 | 87 | 349.37 ± 108.72 | 407.13 | -0.11 (-0.41 to 0.20) | 0.49 |

Baseline values of the questionnaires between Energy conservation and Traditional PR were similar (all P values >0.05).

*All P values <0.01 for change before vs. after for both Energy Conservation and Traditional PR, except for CAT in Traditional PR, where p = 0.38.

**Cohen's d**: Positive value favors Energy conservation and Negative value favors Traditional PR.

**Abbreviations**: PR = Pulmonary Rehabilitation, CAT = COPD Assessment Tool, CRQ = Chronic Respiratory Questionnaire, MMRC = Modified Medical Research Counsel, 6MWT = Six Minute Walk Test.

**Table 3. Change in self-reported difficulty of activities of daily living in energy conservation–pulmonary rehabilitation.**

| (mean ± SD) | Energy Conservation PR (Session 1, n = 85) | Energy Conservation PR (Session 2, n = 45*) | Energy Conservation PR (Session 3, n = 17*) | P-value on Change from Baseline by Repeated Measures ANOVA |
|---|---|---|---|---|
| Stair climbing | 2.99 ± 1.17 | 2.58 ± 1.50 | 2.23 ± 1.44 | 0.04 |
| Reaching | 2.38 ± 1.39 | 1.87 ± 1.41 | 1.29 ± 1.16 | 0.18 |
| Bending | 2.25 ± 1.34 | 1.60 ± 1. 27 | 1.94 ± 1. 25 | 0.19 |
| Vacuuming | 2.38 ± 1.65 | 1.87 ± 1.66 | 1.76 ± 1.64 | 0.46 |
| Total Score | 9.99 ± 4.33 | 8.13 ± 5.12 | 7.23 ± 4.41 | 0.15 |

Abbreviations: PR = Pulmonary Rehabilitation.

*Many patients who attended the sessions chose not to complete the follow-up surveys.

Table 3 summarized the exploratory analyses of the change in self-reported ADL task difficulty amongst patients that participated in energy conservation PR and answered the questionnaires during the sessions. Only 53% (45/85) answered in session 2 and 20% (17/85) answered in all three sessions. Although improvement in ADL task difficulty were noted throughout the sessions, statistically significant changes were only observed for stair climbing (p = 0.04). Regression analyses did not show changes in ADL difficulty between sessions to significantly differ by COPD status (all p values >0.16).

## Discussion

To our knowledge, our study is the first to investigate the efficacy of adding ADL simulation exercises specific to problematic ADLs within a multidisciplinary PR program. This type of training has only been studied as an isolated form of occupational therapy as an alternative to patients with COPD who did not enroll in PR. Martinsen et al. implemented individualized exercise programs with goal directed resistance training specific to domestic problems identified by the Canadian Occupational Performance Measure hoping to improve outcomes of functional independence. These tailored resistance training exercise protocols were designed to replicate ADLs that patients reported to be difficult which included stair-climbing, walking uphill, vacuuming, making the bed, carrying groceries, and personal care tasks. They concluded that tailored resistance training targeted toward improving ADL ability was just as effective as generic resistance training exercise prescribed as part of traditional PR [8].

Only one study to date has combined occupational therapy sessions that applied energy conservation techniques with traditional PR. Vaes et al. administered once weekly occupational therapy sessions with a traditional exercise program which included both generic resistance training limited to machines, high intensity interval training on an arm cycle ergometer and treadmill walking among patients with COPD diagnosis [9]. The primary outcomes of this study focused on changes in oxygen uptake during the performance of ADL testing before and after completion of PR among other functional outcomes. However, they did not address overall program outcomes of dyspnea or HRQOL. They concluded that patients performed the ADL testing in less time, with less metabolic load, and with fewer symptoms of fatigue and dyspnea during testing. While the improvements in ADL testing translated to improved performance of ADLs, their resistance exercise program was not specifically tailored to address problematic tasks. Inclusion of tailored resistance training, application of energy conservation, and ADL simulation training in combination with traditional PR has not been done in the past. We found that incorporation of ADL simulation training into traditional PR is superior to traditional PR in significantly improving patient's HRQOL measured by CAT score despite

the lack of a significant difference in six-minute walk test. We hypothesize that ADL simulation can significantly impact patient-perceived outcomes of HRQOL due to its targeted training effects on performance of difficult ADL's such as stair-climbing without change in objective measures of functional status. Further studies are needed to formally test these assumptions and find strategies to efficiently improve both HRQOL and functional status in PR without adding extra duration of PR sessions.

Despite the improvements seen in HRQOL, between group differences in dyspnea and fatigue scores as well as six-minute walk test were not observed. It is possible that higher dose or duration of the ADL intervention is needed to effect change in these metrics. For example, there is a large difference in the energy requirement of the tasks in the MMRC in order to advance from one category to the next. It is also likely that the muscular groups trained during ADL simulation did not significantly change the walking distance. In addition, while patients were encouraged to practice resistance training specific to ADLs at home, it is likely that the compliance to these difficult exercises is low without on-site supervision.

While generic resistance training guidelines from the American Thoracic Society and the American Association of Cardiovascular and Pulmonary Rehabilitation have been established, they lack guidance for implementing functional training specific to improving ADL task performance. The American Thoracic Society recommends two to three days per week of moderate intensity resistance training using repetitive lifting of relatively heavy loads as the primary modality. The American Association of Cardiovascular and Pulmonary Rehabilitation lacks specific resistance training guidelines but suggest the use of hand and ankle weights, free weights, machine weights, elastic resistance bands, and bodyweight exercises including stair-climbing and squats using lighter loads with higher repetitions to promote the development of muscular endurance [4]. Both organizations are in agreement that the inclusion of upper extremity resistance training is warranted, however, only the American Association of Cardiovascular and Pulmonary Rehabilitation provides direction that the upper limb activities selected be specific to tasks required for functional living [4]. Provided that there remains a gap in how specific resistance training exercise should be implemented for improvement in functional ADLs, the present study may provide the first evidence that investing in the inclusion of ADL training in PR sessions may improve HRQOL without sacrificing walking distance or dyspnea and fatigue scores.

Our study also has limitations. Given the retrospective design of this study, the possibility of change in temporal trends as potential bias as well as other unmeasured confounding factors that provide alternative explanation of our findings cannot be excluded. For example, we did not have data on home adherence to resistance training or to exercise prescription overall which could have partially explained the observed differences between the groups. In addition, since about one third of the patients did not complete PR in our program, our results are applicable only to patients who were able to complete PR. Future randomized-controlled trials that rigorously evaluate the Energy Conservation PR against Traditional PR to confirm our findings are required.

## Conclusion

Our results indicate that when replacing three monthly sessions of traditional PR with energy conservation and ADL simulation training improved HRQOL outcomes over traditional PR alone. Tailoring patient's individualized resistance training programs to include exercises specific to ADLs that the patient finds to be most difficult and training on energy conservation techniques while performing these tasks may help in these patients' HRQOL outcomes.

## Supporting information

**S1 Table. Comparison of patients who completed the pulmonary rehabilitation versus drop-out.**
(DOCX)

**S2 Table. Comparison of the change from baseline in patient outcomes between energy conservation vs. traditional pulmonary rehabilitation by COPD status.**
(DOCX)

## Author Contributions

**Conceptualization:** Kayla Mahoney.

**Data curation:** Kayla Mahoney, Stacey Papo, Hafiz Imran.

**Formal analysis:** Kayla Mahoney, Hafiz Imran, Wen-Chih Wu.

**Investigation:** Kayla Mahoney, Jacqueline Pierce, Stacey Papo, Samuel Evans, Wen-Chih Wu.

**Methodology:** Kayla Mahoney, Jacqueline Pierce, Stacey Papo, Hafiz Imran, Wen-Chih Wu.

**Supervision:** Samuel Evans, Wen-Chih Wu.

**Validation:** Jacqueline Pierce, Samuel Evans.

**Writing – original draft:** Kayla Mahoney.

**Writing – review & editing:** Samuel Evans, Wen-Chih Wu.

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
