## [Decision Letter · Decision Letter 0]

12 May 2020

PONE-D-20-05739

Efficacy of Adding Activity of Daily Living Simulation Training to Traditional Pulmonary Rehabilitation on Dyspnea and Health-Related Quality-of-life

PLOS ONE

Dear Dr Wen-Chih Wu,

Thank you for submitting your manuscript to PLOS ONE. After careful consideration, we feel that it has merit but does not fully meet PLOS ONE’s publication criteria as it currently stands. Therefore, we invite you to submit a revised version of the manuscript that addresses the points raised during the review process.

We would appreciate receiving your revised manuscript by July 6, 2020. To enhance the reproducibility of your results, we recommend that if applicable you deposit your laboratory protocols in protocols.io, where a protocol can be assigned its own identifier (DOI) such that it can be cited independently in the future. For instructions see: http://journals.plos.org/plosone/s/submission-guidelines#loc-laboratory-protocols

We look forward to receiving your revised manuscript.

Kind regards,

Vinicius Cavalheri, Ph.D.

Academic Editor

PLOS ONE

Journal Requirements:

Reviewers' comments:

Reviewer's Responses to Questions

**Comments to the Author**

1. Is the manuscript technically sound, and do the data support the conclusions?

Reviewer #1: Partly

Reviewer #2: Yes

Reviewer #3: No

2. Has the statistical analysis been performed appropriately and rigorously? 

Reviewer #1: Yes

Reviewer #2: Yes

Reviewer #3: No

3. Have the authors made all data underlying the findings in their manuscript fully available?

Reviewer #1: Yes

Reviewer #2: Yes

Reviewer #3: No

4. Is the manuscript presented in an intelligible fashion and written in standard English?

Reviewer #1: Yes

Reviewer #2: Yes

Reviewer #3: Yes

5. Review Comments to the Author

Reviewer #1: The manuscript called “Efficacy of Adding Activity of Daily Living Simulation Training to Traditional Pulmonary Rehabilitation on Dyspnea and Health-Related Quality-of-life” is very interesting and relevant to Pulmonary Rehabilitation area, but there are some considerations to do. It is reported below.

Abstract: clear.

Introduction: The authors explain the traditional pulmonary rehabilitation, justify the ADLs training needs but do not mention the energy conservation.

What is energy conservation?

Why energy conservation is important in this context?

I’m asking it because “The aim of this study was to determine the impact of incorporation of ADL simulation and energy conservation practice in PR as part of a quality-initiative on dyspnea, fatigue, and health-related-quality-of-life (HRQOL) among PR patients.”

Methods:

The authors must define the acronyms used to describe the groups in the methods and, after that, they can use that on the tables.

The description of ADL simulation and PR Groups is not clear.

Do the baseline questionnaire (Self-Reported Task Difficulty) is validated instrument?

The authors said “Therefore, in addition to the standard 12 weeks of twice weekly Traditional PR exercise and education sessions, ADL simulation intervention occurred in place of one regular Traditional PR session on a monthly basis for three months.”

Was there a specific time to start the ADL simulation training? (example: 2 weeks after started the PR or every 2 weeks in each month)

The Table 2 is not necessary, it could be described as part of the protocol, explaining the exercises. If the authors want to keep the table 2, they have to review the title (Table 2. Exercises Included in Energy Conservation Intervention) Where the Energy conservations are there?

When and how the energy conservation was used?

Results:

How many records was assessed?

How many was excluded and why?

Table 1 the result of 6MWT distance is showed in ft. The usual way to report this measure is in meters.

Why is not showed the p value for FEV1 and FEV1/FVC?

Table 3 the authors need to review the CRQF’s data the difference between EC and TPR groups is too big.

Table 4 - Although the p value was significant for Stairclimbing and Reaching, the standard deviations are very large, I recommend reviewing the statistic.

The authors should explain at the end of each table the definition of all the acronyms used.

Discussion:

There are papers published about the results of PR on ADLs and about energy conservation could help the authors discuss their results strongly.

Reviewer #2: This retrospective and observational study aimed to investigate the efficacy of a traditional PR added by an ADL simulation protocol on outcomes such as dyspnea, HRQOL and six-minute walk distance in subjects with chronic respiratory diseases. This is an interesting proposal since this topic of adding ADL simulation exercises to problematic ADLs in a PR program is innovative and it can contribute to the clinical practice. The manuscript is well written and the methodology is well organized. However, there are minor comments and some suggestions in order to improve the quality of the study.

Comments:

- This study reviewed medical records of a miscellaneous of respiratory diseases. In the authors’ opinion , may this study profile interfere in the results? How was the outcome differences of COPD and non-COPD subjects? Maybe this issue could be clearer in the article.

- The tables in the article need to be self-explanatory. Hence, the authors should describe the definition of each abbreviation in the tables 1, 3 and 4.

- The differences found on CAT and CRQ dyspnea are statistically significant, however, are they clinically relevant?

- The baseline values of the six-minute walk distance are mismatched in table 1 and 3. These values should be revised.

- Table 1 shows the baseline characteristics of the subjects between groups as well as the differences of those characteristics. There was any difference in the baseline scores of questionnaires of HRQOL? Maybe this information should be added in this table.

- Along the text of the manuscript the table 2 is mentioned prior to table 1. Maybe this citation need to be revised.

Reviewer #3: The study of Mahoney and colleagues investigated the addition of an ADL simulation protocol to PR on clinical outcomes in a variety of pulmonary diseases. The study was done collecting data retrospectively from an outpatient clinic. It is not clear, though, whether the data is from previous investigations or whether this is a real-life scenario. Authors conclude that the protocol provided additional benefits on quality of life and dyspnea. This is not, however, supported by the statistical treatment. Additionally, the temporal bias of the data collection was never mentioned in the manuscript. Specific comments are provided below.

Page 4. The heterogeneity of the sample is worrisome. Patients with different ILD respond different to treatment and they all differ from COPD. It is difficult to know whether the training responses observed (or the lack thereof) are due to the protocol or due to the large variety of diseases. I recommend choosing only one disease (e.g. COPD) to be reported.

Page 4. “ For this study, we only reviewed the records of patients who completed the 12 weeks of the PR program, which was 62% in 2017 and 58% in 2018.” Despite the retrospective design, it is a major limitation not having the data of attrition. If available, please include baseline characteristics of all patients (including dropouts) and describe reasons for non-completion of the protocol.

Page 5. As per definition, primary outcome is only one. Authors report three (MMRC, CRQ and CAT).

Page 7. “… between group differences in primary and secondary outcomes were compared using paired-sample and two-sample T tests respectively.” What exactly was compared between groups? Changes (Δ) or the assessments at week 12? A 2-way ANOVA seems more appropriate. The magnitude of differences (e.g. Cohen´s d test) is needed. Finally, it is recommended to report whether observed differences in changes are clinically relevant (i.e. exceeding MIDs of the tests).

Page 9. “We found that incorporation of ADL simulation training into Traditional PR is superior to Traditional PR in significantly improving patient’s HRQOL measured by CAT score and CRQ Dyspnea score despite the lack of a significant difference in six-minute walk test.” This is not (yet) a conclusion of your study. Authors found significant differences in the ADL group. This, however, does not necessarily means there is superiority as statistical treatment is inadequate.

Page 10. “ In addition, while patients were encouraged to practice resistance training specific to ADLs at home, it is likely that the compliance to these difficult exercises is low without on-site supervision.”. This remains controversial in the literature. Have you controlled it?

6. PLOS authors have the option to publish the peer review history of their article (what does this mean?). If published, this will include your full peer review and any attached files.

Reviewer #1: No

Reviewer #2: No

Reviewer #3: No

---

## [Author Response · Author response to Decision Letter 0]

14 Jul 2020

July 12, 2010

PONE-D-20-05739

Efficacy of Adding Activity of Daily Living Simulation Training to Traditional Pulmonary Rehabilitation on Dyspnea and Health-Related Quality-of-life

PLOS ONE

Dear Editor and Reviewers,

Thank you for the opportunity to allow us to improve the manuscript. A draft with track changes in the manuscript was also submitted along with a clean version. Please see below an item to item response to the reviewer’s comments in red.

Reviewer #1: The manuscript called “Efficacy of Adding Activity of Daily Living Simulation Training to Traditional Pulmonary Rehabilitation on Dyspnea and Health-Related Quality-of-life” is very interesting and relevant to Pulmonary Rehabilitation area, but there are some considerations to do. It is reported below.

Abstract: clear.

Introduction: The authors explain the traditional pulmonary rehabilitation, justify the ADLs training needs but do not mention the energy conservation.

1. What is energy conservation? Why energy conservation is important in this context?

I’m asking it because “The aim of this study was to determine the impact of incorporation of ADL simulation and energy conservation practice in PR as part of a quality-initiative on dyspnea, fatigue, and health-related-quality-of-life (HRQOL) among PR patients.”

Energy conservation and its importance is now defined in the introduction and AACVPR guidelines for the inclusion of this technique in all PR programs is now referenced (p.3 last paragraph, p.4 first 2 paragraphs).

Methods:

2. The authors must define the acronyms used to describe the groups in the methods and, after that, they can use that on the tables.

Acronyms are now defined in methods in the subtitle of “Comparison Groups” and tables. We now minimized the use of acronyms to improve readability. We now used “Traditional Pulmonary Rehabilitation (Traditional PR)” for one group and “Energy Conservation Pulmonary Rehabilitation (Energy Conservation PR)” for the other group. 

3. The description of ADL simulation and PR Groups is not clear. 

We reorganized the methods section for clarity and added further details to the description of Energy conservation in ADL simulation and the distinction of both PR groups (p. 7 last paragraph, pp. 8-9).

4. Do the baseline questionnaire (Self-Reported Task Difficulty) is validated instrument? 

The Self-Reported Task Difficulty questionnaire was adopted from a portion of the CAT and is not yet validated amongst the non-COPD population. This is now clarified in the methods section of the manuscript (p.8, last paragraph).

5. The authors said “Therefore, in addition to the standard 12 weeks of twice weekly Traditional PR exercise and education sessions, ADL simulation intervention occurred in place of one regular Traditional PR session on a monthly basis for three months.”

Was there a specific time to start the ADL simulation training? (example: 2 weeks after started the PR or every 2 weeks in each month) 

There was no restriction on when patients can start their ADL simulation training in Pulmonary Rehab. The Pulmonary rehab program offered ADL simulation intervention in place of a regularly scheduled Traditional Pulmonary Rehab session on the second or third Tuesday of every month over the course of the year (Sept 2017-Sept 2018). Thus, patients would have up to three separate occasions to receive energy conservation on ADL simulation training during their 12-week session. This is now clarified in the methods section (p.8, second paragraph).

6. The Table 2 is not necessary, it could be described as part of the protocol, explaining the exercises. If the authors want to keep the table 2, they have to review the title (Table 2. Exercises Included in Energy Conservation Intervention) Where the Energy conservations are there?

When and how the energy conservation was used?

Table 2 has been removed and exercises have been described in detail in the methods section (pp. 8-9). In addition, we added further details in the paragraph of when and how the energy conservation was used. 

Results:

7. How many records was assessed? 

Medical records of 176 patients who completed the 12-week PR program were reviewed in the study and analyzed. This was clarified in the first sentence of the results section (p.11).

8. How many was excluded and why? 

Patients who did not complete the PR program were excluded since we would not have post-PR questionnaires and assessments for comparison. We have provided the number of patients that were enrolled in the program for each group, in addition to the # of patients that dropped out. We have also included the top 3 reasons for drop out (p. 5, last paragraph). 

9. Table 1 the result of 6MWT distance is showed in ft. The usual way to report this measure is in meters. 

Converted feet to meters in Table 1.

10. Why is not showed the p value for FEV1 and FEV1/FVC? 

We apologize for the typo, P values for FEV1 and FEV1/FVC have now been added

11. Table 3 the authors need to review the CRQF’s data the difference between EC and TPR groups is too big.

We apologize for the typo, the values for CRQF for Energy Conservation PR, pre and post, have been revised.

12. Table 4 - Although the p value was significant for Stairclimbing and Reaching, the standard deviations are very large, I recommend reviewing the statistic.

We thank the reviewer for the astute observation. We apologize for the typo. Per reviewer 3’s suggestion, we have now changed this to an exploratory analysis given significant missing and non-validated questionnaire, reported the available data on all three visits, and used repeated measures ANOVA for analysis. Only Stair climbing remained significant. The old Table 4, is now new Table 3.

13. The authors should explain at the end of each table the definition of all the acronyms used.

Thank you for the excellent suggestion. We have added the full wording of the acronyms as a footnote on each table.

Discussion:

14. There are papers published about the results of PR on ADLs and about energy conservation could help the authors discuss their results strongly. 

Given the novelty of the idea in pulmonary rehab, there is a paucity of data in the topic. However, we have included now 4 papers in the topic including a paper published in 2019 on incorporation of energy conservation techniques in pulmonary rehab and its effect on functional performance measured by six-minute walk test and ADL task testing in the discussion (p.12, last paragraph, p.13, first paragraph). 

Reviewer #2: This retrospective and observational study aimed to investigate the efficacy of a traditional PR added by an ADL simulation protocol on outcomes such as dyspnea, HRQOL and six-minute walk distance in subjects with chronic respiratory diseases. This is an interesting proposal since this topic of adding ADL simulation exercises to problematic ADLs in a PR program is innovative and it can contribute to the clinical practice. The manuscript is well written and the methodology is well organized. However, there are minor comments and some suggestions in order to improve the quality of the study.

Comments:

15. This study reviewed medical records of a miscellaneous of respiratory diseases. In the authors’ opinion, may this study profile interfere in the results? How was the outcome differences of COPD and non-COPD subjects? Maybe this issue could be clearer in the article.

We thank the reviewer for the excellent suggestion. We added a test for multiplicative interaction between type of PR and COPD diagnosis using linear regression to assess whether study findings varied by COPD status (p. 10, second paragraph). Results were summarized in the results section (last 2 paragraphs p.11 and first paragraph p.12). 

16. The tables in the article need to be self-explanatory. Hence, the authors should describe the definition of each abbreviation in the tables 1, 3 and 4.

Footnote has been added on each table to describe the abbreviations

17. The differences found on CAT and CRQ dyspnea are statistically significant, however, are they clinically relevant?

We have added in the methods section the point change after which CAT and CRQ dyspnea scores are considered significant. Based on the recommendations by the American Association of Cardiovascular and Pulmonary Rehabilitation guidelines, a change ≥2 points in the negative direction on the CAT indicates less impact of COPD severity on a patients HRQOL (p.6, second paragraph). Improvement in CRQ-D scores ≥ 0.5 point are considered clinically relevant based on American Association of Cardiovascular and Pulmonary Rehabilitation guidelines (p.6, third paragraph, p.7 first paragraph). 

18. The baseline values of the six-minute walk distance are mismatched in table 1 and 3. These values should be revised.

We apologize for the error, change has been made and reflects distance in meters instead of feet. Tables are now consistent pre and post.

19. Table 1 shows the baseline characteristics of the subjects between groups as well as the differences of those characteristics. There was any difference in the baseline scores of questionnaires of HRQOL? Maybe this information should be added in this table.

Baseline values of the questionnaires between Energy conservation and Traditional PR were similar (all P values >0.05). This is addressed in the results section (p.11, second paragraph) and footnote in new Table 2.

20. Along the text of the manuscript the table 2 is mentioned prior to table 1. Maybe this citation need to be revised.

This has been revised. Table 1 mentioned prior to the new Table 2 in the results section.

Reviewer #3: The study of Mahoney and colleagues investigated the addition of an ADL simulation protocol to PR on clinical outcomes in a variety of pulmonary diseases. The study was done collecting data retrospectively from an outpatient clinic. 

21. It is not clear, though, whether the data is from previous investigations or whether this is a real-life scenario. 

This is data collected from a real-life scenario. To clarify further, we modified the following sentence under study design (p. 5, first paragraph): “The study was a retrospective, observational comparison of patients enrolled in PR before (October 2016 through August 2017) and after (September 2017 to September 2018) the addition of the new energy conservation in ADL simulation training to Traditional PR intervention, as part of an institutional quality-improvement initiative in September of 2017”. In addition, in methods under study sample we stated (p. 5, second paragraph): “Medical records of 176 patients with qualifying diagnoses for PR … who completed PR in either a University Teaching Hospital or a Community Hospital were reviewed.”

22. Authors conclude that the protocol provided additional benefits on quality of life and dyspnea. This is not, however, supported by the statistical treatment. Additionally, the temporal bias of the data collection was never mentioned in the manuscript. Specific comments are provided below.

We have made modifications in the statistical methods using repeated measures ANOVA as suggested (p. 10, second paragraph) and added in the limitation section of the discussion “Given the retrospective design of this study, the possibility of change in temporal trends as potential bias as well as other unmeasured confounding factors that provide alternative explanation of our findings cannot be excluded” (p. 14, second paragraph). 

23. Page 4. The heterogeneity of the sample is worrisome. Patients with different ILD respond different to treatment and they all differ from COPD. It is difficult to know whether the training responses observed (or the lack thereof) are due to the protocol or due to the large variety of diseases. I recommend choosing only one disease (e.g. COPD) to be reported.

To address this excellent point and per the suggestion of reviewer 1, we made the following modifications in statistical analysis (p.10, second paragraph): “Given that PR encompassed a heterogeneous group of patients, to assess whether study findings vary by COPD status, we tested for multiplicative interaction between type of PR and COPD diagnosis using linear regression where the dependent variable was change from baseline for each of the study outcomes.” Results were summarized in the results section (p.11, second paragraph): “Regression analyses using multiplicative interaction did not show any of the above results to significantly differed by COPD status (all p values >0.25)”. 

24. Page 4. “For this study, we only reviewed the records of patients who completed the 12 weeks of the PR program, which was 62% in 2017 and 58% in 2018.” Despite the retrospective design, it is a major limitation not having the data of attrition. If available, please include baseline characteristics of all patients (including dropouts) and describe reasons for non-completion of the protocol.

We reviewed and compared baseline characteristics (age, gender and COPD diagnosis) of the patients who completed the program versus those who did not and summarized them in a new Supplemental Table and described them in methods (p.5, last paragraph). We also described the top three reasons for non-completion amongst the dropped-out population in the same paragraph. 

25. Page 5. As per definition, primary outcome is only one. Authors report three (MMRC, CRQ and CAT).

We reorganized the text in the methods section to reflect that the primary outcome of the study is HRQOL, as measured by CAT, since it encompassed a spectrum of domains that determined a patient’s holistic self-sense of well-being and endorsed by both American Thoracic Society and the American Association of Cardiovascular and Pulmonary Rehabilitation (p.6, second paragraph). 

26. Page 7. “… between group differences in primary and secondary outcomes were compared using paired-sample and two-sample T tests respectively.” What exactly was compared between groups? Changes (Δ) or the assessments at week 12? A 2-way ANOVA seems more appropriate. The magnitude of differences (e.g. Cohen´s d test) is needed. 

We apologize for lack of clarity. Per the reviewer’s suggestions we have now compared the change from baseline within the group and between the groups in primary and secondary outcomes using repeated measures ANOVA and calculated the Cohen’s d for the between group differences for each study outcome (p. 9, second to last sentence).

27. Finally, it is recommended to report whether observed differences in changes are clinically relevant (i.e. exceeding MIDs of the tests).

We have added in the methods section the point change after which CAT and CRQ scores (dyspnea and fatigue) were considered clinically significant. Based on the recommendations by the American Association of Cardiovascular and Pulmonary Rehabilitation, a change of ≥2 points in the negative direction on the CAT indicates less impact of COPD severity on a patients HRQOL (p.6, third paragraph) and an improvement in CRQ-D scores of ≥0.5 point are considered clinically relevant (p.6, last paragraph). 

27. Page 9. “We found that incorporation of ADL simulation training into Traditional PR is superior to Traditional PR in significantly improving patient’s HRQOL measured by CAT score and CRQ Dyspnea score despite the lack of a significant difference in six-minute walk test.” This is not (yet) a conclusion of your study. Authors found significant differences in the ADL group. This, however, does not necessarily means there is superiority as statistical treatment is inadequate.

We have carefully redone the statistical analyses using repeated measures ANOVA as suggested by the reviewer have tone down our conclusions to reflect that a randomized-controlled trial will be needed to confirm these initial findings (p.14 last paragraph, p. 15).

28. Page 10. “ In addition, while patients were encouraged to practice resistance training specific to ADLs at home, it is likely that the compliance to these difficult exercises is low without on-site supervision.”. This remains controversial in the literature. Have you controlled it?

Similar to other observational studies in pulmonary rehab, we do not have accurate data on home adherence to resistance training or home adherence to exercise overall. We will include this as a limitation: “we did not have data on home adherence to resistance training or to exercise prescription overall which could have partially explained the observed differences between the groups” (p. 14, last paragraph). 

We are grateful for the insightful comments that made this a well-rounded, higher quality manuscript.

Sincerely,

Kayla Mahoney, MS & Wen-Chih Wu, MD

---

## [Decision Letter · Decision Letter 1]

4 Aug 2020

PONE-D-20-05739R1

Efficacy of Adding Activity of Daily Living Simulation Training to Traditional Pulmonary Rehabilitation on Dyspnea and Health-Related Quality-of-life

PLOS ONE

Dear Dr. Wu,

Thank you for submitting your manuscript to PLOS ONE. After careful consideration, we feel that it has merit but does not fully meet PLOS ONE’s publication criteria as it currently stands. Therefore, we invite you to submit a revised version of the manuscript that addresses the points raised during the review process.

We look forward to receiving your revised manuscript.

Kind regards,

Vinicius Cavalheri, Ph.D., MSc

Academic Editor

PLOS ONE

Additional Editor Comments (if provided):

Dear Dr Wen-Chih Wu,

Thank you for addressing the comments from the Reviewers.

There are only two minor changes suggested by Reviewer #1. Can I please ask you to address the two minor changes?

Many thanks

Reviewers' comments:

Reviewer's Responses to Questions

**Comments to the Author**

1. If the authors have adequately addressed your comments raised in a previous round of review and you feel that this manuscript is now acceptable for publication, you may indicate that here to bypass the “Comments to the Author” section, enter your conflict of interest statement in the “Confidential to Editor” section, and submit your "Accept" recommendation.

Reviewer #1: (No Response)

Reviewer #3: All comments have been addressed

2. Is the manuscript technically sound, and do the data support the conclusions?

Reviewer #1: Yes

Reviewer #3: Yes

3. Has the statistical analysis been performed appropriately and rigorously? 

Reviewer #1: Yes

Reviewer #3: Yes

4. Have the authors made all data underlying the findings in their manuscript fully available?

Reviewer #1: Yes

Reviewer #3: No

5. Is the manuscript presented in an intelligible fashion and written in standard English?

Reviewer #1: Yes

Reviewer #3: Yes

6. Review Comments to the Author

Reviewer #1: The manuscript called “Efficacy of Adding Activity of Daily Living Simulation Training to Traditional Pulmonary Rehabilitation on Dyspnea and Health-Related Quality-of-life” improved after revision but remain some details that needs the authors attention.

Methods:

Pag 5-6 – “…Out of the 142 patients enrolled between October 2016 and August 2017 (pre-implementation period), 51 (35.9%) patients…, These patients…and personal reasons.” It is Results, not Methods. I suggest the authors transfer that paragraph to the beginning of results.

Suggestion to reduce the study limitations – The authors said “The questionnaires used to assess HRQOL,15 dyspnea and fatigue were not validated for non-COPD patients”

Looking the table 1 the authors has 59 COPD patients on EC-PR group and 54 on TPR group, the others chronic lung disease together adds 46 individuals in each group. The authors might do the separately analyzes of data COPD and not COPD. Perhaps the authors could do a separate analysis of COPD and non-COPD data. In my opinion, this will make the result stronger.

Reviewer #3: The authors have made substantial improvement a to the manuscript. I believe the manuscript is now suitable for publication.

7. PLOS authors have the option to publish the peer review history of their article (what does this mean?). If published, this will include your full peer review and any attached files.

Reviewer #1: No

Reviewer #3: No

---

## [Author Response · Author response to Decision Letter 1]

4 Aug 2020

August 4, 2020

PONE-D-20-05739

Efficacy of Adding Activity of Daily Living Simulation Training to Traditional Pulmonary Rehabilitation on Dyspnea and Health-Related Quality-of-life

PLOS ONE

Dear Editor and Reviewers,

Thank you for the opportunity to allow us to improve the manuscript. A draft with track changes in the manuscript was also submitted along with a clean version. Please see below an item to item response to the reviewer’s comments in red.

Reviewer #1: The manuscript called “Efficacy of Adding Activity of Daily Living Simulation Training to Traditional Pulmonary Rehabilitation on Dyspnea and Health-Related Quality-of-life” improved after revision but remain some details that needs the authors attention.

1. Methods:

Pag 5-6 – “…Out of the 142 patients enrolled between October 2016 and August 2017 (pre-implementation period), 51 (35.9%) patients…, These patients…and personal reasons.” It is Results, not Methods. I suggest the authors transfer that paragraph to the beginning of results.

Paragraph has been transferred to the beginning of the results.

2. Suggestion to reduce the study limitations – The authors said “The questionnaires used to assess HRQOL,15 dyspnea and fatigue were not validated for non-COPD patients”. Looking the table 1 the authors has 59 COPD patients on EC-PR group and 54 on TPR group, the others chronic lung disease together adds 46 individuals in each group. The authors might do the separately analyzes of data COPD and not COPD. Perhaps the authors could do a separate analysis of COPD and non-COPD data. In my opinion, this will make the result stronger.

A subgroup analysis on the patient self-reported outcomes by COPD status was added as a supplemental table 2 and to results: “Subgroup analysis by COPD status was described in Supplemental Table 2. The change from baseline in CAT scores was significantly greater for the Energy Conservation versus Traditional PR in the COPD group (p=0.03) and trended towards significance in the non-COPD group (p=0.06). The change from baseline in the remaining outcomes (CRQ-dyspnea, CRQ-Fatigue, MMRC, six-minute-walk test) was similar between the Energy Conservation and the Traditional PR groups in both the COPD and non-COPD patients.” In the results section, the regression analysis does not show that the Patient outcomes significantly changed by COPD status. This was added. “Regression analyses using multiplicative interaction did not show any of the above results to significantly differ by COPD status (all P values >0.25)” (last paragraph, p. 11 and first paragraph, p. 12).

As result, we also deleted the sentence from limitations of “The questionnaires used to assess HRQOL, dyspnea and fatigue were not validated for non-COPD patients”. 

We are grateful for the insightful comments that made this a higher quality manuscript.

Sincerely,

Kayla Mahoney, MS & Wen-Chih Wu, MD

---

## [Editor Report · Decision Letter 2]

7 Aug 2020

Efficacy of Adding Activity of Daily Living Simulation Training to Traditional Pulmonary Rehabilitation on Dyspnea and Health-Related Quality-of-life

PONE-D-20-05739R2

Dear Dr. Wu,

We’re pleased to inform you that your manuscript has been judged scientifically suitable for publication and will be formally accepted for publication once it meets all outstanding technical requirements.

Kind regards,

Vinicius Cavalheri, Ph.D., MSc, BSc (PT)

Academic Editor

PLOS ONE

---

## [Editor Report · Acceptance letter]

12 Aug 2020

PONE-D-20-05739R2 

Efficacy of Adding Activity of Daily Living Simulation Training to Traditional Pulmonary Rehabilitation on Dyspnea and Health-Related Quality-of-life 

Dear Dr. Wu:

I'm pleased to inform you that your manuscript has been deemed suitable for publication in PLOS ONE. Congratulations! Your manuscript is now with our production department. 

Kind regards, 

on behalf of

Dr. Vinicius Cavalheri 

Academic Editor

PLOS ONE